# Extracellular Vesicles in the Blood of Dogs with Cancer—A Preliminary Study

**DOI:** 10.3390/ani9080575

**Published:** 2019-08-19

**Authors:** Magdalena Żmigrodzka, Olga Witkowska-Piłaszewicz, Alicja Rzepecka, Anna Cywińska, Dariusz Jagielski, Anna Winnicka

**Affiliations:** 1Department of Pathology and Veterinary Diagnostics, Faculty of Veterinary Medicine, Warsaw University of Life Sciences (WULS-SGGW), Nowoursynowska 159c, 02-787 Warsaw, Poland; 2Veterinary Clinic BIALOBRZESKA, Częstochowska 20, 02-344 Warsaw, Poland

**Keywords:** canine, flow cytometer, platelet microparticles, cancer

## Abstract

**Simple Summary:**

Extracellular vesicles are a diverse population of submicron-sized structures released from cells under physiological and pathological conditions. Due to their size, their role in cell-to-cell communication in cancer is currently being discussed. In blood, the most abundant population is platelet-derived extracellular vesicles. The aim of this study was to estimate the absolute number and the origin of extracellular vesicles in the blood of healthy dogs and of dogs with various types of cancer. The number of extracellular vesicles derived from platelets, leukocytes, and T lymphocytes was significantly higher in dogs with cancer compared to healthy controls. Estimation of platelet-derived extracellular vesicle (PEV) and leukocyte-derived EV counts may provide a useful biological marker in dogs with cancer.

**Abstract:**

Extracellular vesicles (EVs) are a heterogeneous population of submicron-sized structures released during the activation, proliferation, or apoptosis of various types of cells. Due to their size, their role in cell-to-cell communication in cancer is currently being discussed. In blood, the most abundant population of EVs is platelet-derived EVs (PEVs). The aim of this study was to estimate the absolute number and the origin of EVs in the blood of healthy dogs and of dogs with various types of cancer. The EV absolute number and cellular origin were examined by flow cytometry technique. EVs were classified on the basis of surface annexin V expression (phosphatidylserine PS+) and co-expression of specific cellular markers (CD61, CD45, CD3, CD21). The number of PEVs was significantly higher in dogs with cancer (median: 409/µL, range: 42–2748/µL vs. median: 170/µL, range: 101–449/µL in controls). The numbers of EVs derived from leukocytes (control median: 86/µL, range: 40–240/µL; cancer median: 443/µL, range: 44–3 352/µL) and T cells (control median: 5/µL, range: 2–66/µL; cancer median: 108/µL, range: 3–1735/µL) were higher in dogs with neoplasia compared to healthy controls. The estimation of PEV and leukocyte-derived EV counts may provide a useful biological marker in dogs with cancer.

## 1. Introduction

Microparticles were reported in human blood for the first time by Peter Wolf in 1967 and named “platelet dust” [1,2]. Since that time, they have been intensively examined in blood, and initially were classified as cell membrane fragments or other apoptotic bodies. However, this point of view has been questioned in the last two decades, and it has now been proven that they are formed during numerous physiological and pathological conditions [3].

A wide variety of eukaryotic cells release spherical membrane submicron structures into the extracellular space during their activation, proliferation, or in response to hypoxia [1,3]. These subcellular structures, named extracellular vesicles (EVs), are classified into two groups on the basis of their size: exosomes (EXSMs)—50–150 nm and ectosomes (ECTSMs)—150nm–1 µm [2,3]. EVs have been detected in peripheral blood, urine, and other body fluids.

The formation of ECTSMs was described as “blebbing” from the cell membrane [4]. Calcium ions released from the endoplasmic reticulum activate floppase and scramblase and inactivate cell membrane flippase, which leads to exchange of phosphatidylserine (PS) from the inner to the outer side of the cell membrane. Calcium ions also activate gelsolin and calpanin and partially degrade actin filaments, cleaving their contact with phospholipids, which initiates ECTSM budding [2,4,5].

EXSM formation starts with membrane bulging of endosome into the lumen, which results in the formation of a multivesicular body (MVB). Rab family proteins then assist in MVB transport and fusion with cell membrane. Transmembrane protein complex SNARE enables EXSM docking with the cell membrane, and they are then released into extracellular space [6].

EVs contain some parental glycoproteins and surface receptors: CD61 (platelets), CD235a (erythrocytes), PS, and tetraspanins: CD9, CD63, and MHCII molecules. Their composition of mRNA, miRNA, heat shock proteins, matrix metalloproteinases, and cytoskeletal proteins depends to some extent on the cell type, and can be influenced by parental cell conditions [1,3,7]. EVs are heterogeneous in their cellular origin and cargo. There is an increasing body of evidence suggesting that these nanosized vesicles, transported in body fluids for a long or short distance, may be messengers in cell-to-cell communication [1,8]. EVs use various mechanisms to transfer biological information. After fusion with the target cell, parental receptors and adhesive molecules are transferred, and the function of the donor cells is changed due to shifting ligands and production of cytokines and growth factors. Moreover, horizontally transferred mRNA can cause the epigenetic reprogramming of target cells [9]. 

Elevated levels of circulating EVs have been reported in many pathological conditions, e.g., cancers, sepsis, chronic renal failure, or stroke [10]. In the blood stream, 70–90% of EVs originate from platelets (PEVs), followed by EVs from leukocytes and endothelial cells [2,9].

In human breast and colon cancers, increased PEV number has been reported as an unfavorable prognostic factor. Many studies in vitro and in vivo have confirmed that PEVs promote cancer progression and metastasis [10,11,12,13,14]. However, according to the authors’ best knowledge, there is no information regarding EVs in canine cancers. Thus, the aim of this study was to measure the number of PEVs and EVs from leukocytes and T and B lymphocytes in the blood of dogs with cancer.

## 2. Materials and Methods

### 2.1. Animals

Thirteen healthy dogs (control group) and 16 dogs with newly diagnosed cancer (examined group) were included in the study. In the examined group, six dogs had diagnosis of mammary adenocarcinoma, two had T-cell lymphoma, and the remaining cases (one dog each) were acute lymphoblastic leukemia, melanoma, testis adenocarcinoma, uterus sarcoma, liver adenocarcinoma, angiosarcoma, lipoma, and diffuse large B-cell lymphoma. All dogs from the studied group were classified to surgical treatment and/or chemotherapy, based on type of the neoplasia and its clinical stage. All individuals from the examined group, except the dog with lipoma, died after/without treatment within 7 months of diagnosis.

### 2.2. Sample Collection and EV Isolation

Two milliliter peripheral blood samples left after routine veterinary diagnostic procedures were used for the evaluation of blood EVs. The samples from control dogs were taken diagnostically before planned surgery (castration) or periodic laboratory tests. Thus, the research was not subject to Polish regulations concerning animal experiments and did not require the approval of the local ethics committee. 

Blood samples were non-traumatically collected from peripheral veins (cephalic or lateral saphenous) with 21 gauge (G) needles into 4 mL vacutainer tubes spray-coated with sodium heparin. Samples were kept vertically at 4 °C before the next procedures, performed within 30 min of sampling. The inclusion criteria for the healthy control group were: no signs or history of neoplastic disease, lack of clinical and laboratory signs of the disease. Additional criteria included nonsteroidal anti-inflammatory drug (NSAID) treatment or vaccination for two weeks before sampling. The examined group consisted of dogs with cancer but without NSAID therapy up to 2 weeks before blood collection.

The preparation of platelet-free plasma (PFP) consisted of one-step centrifugation for 15 min at 3000× *g* (MPV-351R; MPW med. instruments, Warsaw, Poland) at room temperature (RT). The upper half (an aliquot of 500 µL) of the supernatant (the PFP) was carefully pipetted, frozen immediately, and stored at −20 °C for subsequent analysis for no longer than 4 months. PFPs were subsequently thawed and centrifuged for 15 min at 3000× *g* at RT. Measures of 200 µL of PFP were carefully transferred into cytometric tubes for the next procedures. 

### 2.3. Flow Cytometry and EV Enumeration

Buffers, staining solutions, and FACS flow fluid used in study were filtered through 0.1 µm membrane filters (Corning, NY, USA) [14]. All antibodies (Ab) and isotype controls were centrifuged for 5 min at 20,000× *g* (Becton Dickinson-BD, Franklin Lakes, NJ, USA; Bio-rad, Hercules, CA, USA) [15]. All monoclonal antibodies were species-specific except CD61, which has a documented cross reactivity [16]. Aliquots of 20 µL PFP were added into sterile cytometric tubes and labeled with one of the combinations of antibodies or isotype controls. The following antibody combinations were used: (1) none, (2) annexin V control (annexin V:PE, BD; annexin V:FITC, BD), (3) isotype control CD61/CD45 (mIgG1κ:PE, BD; rIgG2b:APC, Bio-rad), (4) isotype control CD3/CD21 (mIgG1:FITC, Bio-rad; mIgG1:AF647, Bio-rad), (5) CD61:PE/CD45:APC (clon VI-PL2, BD; clon YKIX718.13, Bio-rad), (6) CD3:FITC/CD21: AF647 (clone CA17.2A12, Bio-rad; clone CA2.1D6, Bio-rad). PS expression on EVs and control of annexin V was measured using FITC- or PE-labeled annexin V (annexin V:FITC or PE, BD) added into tubes 2, 5, and 6. Samples were incubated for 20 min at RT in the dark, and then 450 µL of annexin V buffer was added into tubes 5 and 6 and 450 µL of DPBS into remaining tubes. Samples then were transferred into TruCountTM (BD) tubes with lyophilized pellet, with a known number of fluorescent beads, and analyzed within 1 h in a flow cytometer FACSCanto II (BD). All data analysis was carried out by Cell Quest software.

The absolute number of selected microparticles per microliter of sample (EVs/µL) was calculated by the following formula: *EVs*/μL = (G_Mv_ × TC)/(G_TC_ × V)(1)
where G_MV_—number of events in region containing EVs, TC—number of beads per test, G_TC_—number of events in absolute count bead region, and V—test volume [14,17]. The number of beads per test tube (TruCountTM tube) was provided by the manufacturer for every tube, and the test volume was 20 µL.

The critical element of the standardization protocol is to use of appropriate reference beads. The EV gate was established on a FACSCanto II cytometer by standardization experiment using size-calibrated beads, SSC Megamix beads (Biocytex, Marseille, France), recommended for this type of instrument [15]. Using the SSC-H scale to create EV regions instead of the FSC-H scale was proven by Poncelet at al. to be an optimal method for EV detection for that type of flow cytometer, and is comparable to the FSC–H scale for 0.3–1 µm calibrating beads [15]. An EV gating strategy based on size-calibrated beads ranging from 0.16 µm to 0.5 µm was created, as shown in Figure 1A. The upper size limit for EVs was defined using a side-scatter (SSC-H) gate on 0.5 µm, and the lowest on 0.20 µm. The instrument settings and threshold on the FITC eliminated the smallest (0.16 µm) beads and permitted creation of the gate for the TruCountTM beads (Figure 1B).

In the EV gate, the following populations were classified: PEVs, leukocyte EVs, B-cell EVs, and T-cells EVs, based on PS expression using annexin V with co-expression of: CD61—PEVs, CD45—leukocyte-derived EVs, CD21—B-cell-derived EVs, and CD3—T-cell-derived EVs. Figure 2 shows exemplary dot plots of the EV phenotyping in healthy dogs and in dogs with cancer.

### 2.4. Statistical Analysis

Statistical significance of the differences between the groups was assessed using nonparametric Mann–Whitney U tests. All data are shown as median with ranges. Sperman’s rho correlation test was conducted. All statistical analyses were performed using Statistica 13 (TIBCO Software, Palo Alto, CA, USA). For all analyses, *p*-values < 0.05 were considered significant.

## 3. Results

### 3.1. Study Population

Due to the fact that clinical patients were examined, the group was heterogeneous. The dogs were between 9 months and 13 years old, but there was no significant difference in age between control dogs and dogs with cancer (control median: 3.0, range: 0.75–8.5 years; cancer median: 8.0, range: 5–13 years; *p* > 0.05). In the control group, there were three spayed females, five intact females, four intact males, and one neutered male. In the examined group, there were one neutered male, three spayed females, seven intact females, and four intact males. There were two Bernese mountain dogs, nine mixed-breed, and one of each Labrador, dogue de Bordeaux, Rottweiler, boxer, and Siberian husky in the examined group. The control group consisted of seven mixed-breed dogs, three border collies and one of each German shepherd, Cavalier King Charles spaniel, and boxer.

### 3.2. Hematological Variables

Results for complete CBC testing were available for 14 of the 16 dogs with cancer and all 13 control dogs. Only the platelet counts (PLT) were significantly higher in dogs with cancer, WBC counts were higher in the examined group, which may indicate a trend, although the significance has not yet been proven. There were no differences between the groups for red blood cell count, mean corpuscular volume (MCV) and mean corpuscular hemoglobin concentration (MCHC), and hematocrit (Table 1).

### 3.3. Flow Cytometry Assessment of Blood Circulating EV in Dogs with Cancer

PEVs were evaluated by flow cytometry in all 13 healthy and 15 of 16 dogs with cancer. One dog was eliminated from the examined group because of PFP hemolysis (intact female with mammary adenocarcinoma). PEVs were classified as PS+ D61+ events. The median number of PEVs was significantly higher in dogs with cancer versus the control group (control median: 170/µL, range: 101–449/µL; cancer median: 409/µL, range: 42–2748/µL; *p* < 0.05) (Figure 3A). The PEV number was the highest in the dog with testis adenocarcinoma, and second highest in the bitch with uterus sarcoma with metastasis in the bladder. 

The numbers of PS+ CD61+ CD45+ EVs were also counted (control median: 28/µL, range: 4–95/µL; cancer median: 35/µL, range: 2–729/µL) and were found to be higher in dogs with cancer, although without significance. The highest values were noticed in the same two dogs with the highest PEVs numbers.

The leukocyte-derived EVs were defined as PS+ CD45+ events. Their number was significantly higher in dogs with neoplasia compared to the control group (control median: 86/µL, range: 40–240/µL; cancer median: 443/µL, range: 44–3352/µL; *p* < 0.05) (Figure 3B). Again, the highest values of leukocyte-derived EVs as PEVs were noticed in the two dogs mentioned above.

EVs from T cells were identified as PS+ CD3+ microparticles, and were significantly higher in dogs with neoplasia (control median: 5/µL, range: 2–66/µL; cancer median: 108/µL, range: 3–1735/µL; *p* < 0.05) (Figure 3C). The numbers of T-cell EVs were the highest in two dogs with T-cell lymphoma and the dog with diffuse B-cell lymphoma.

The population of EVs from B cells was defined as PS+ CD21+ EVs (B-cell EVs). Again, the number of B-cell EVs was higher in dogs with cancer, compared to healthy controls (control median: 6/µL, range: 0–23/µL; cancer median: 4/µL, range: 0–188/µL). The highest values were observed in the dog with diffuse B-cell lymphoma and the dog with liver adenocarcinoma with lung metastasis.

Box and whisker plots illustrating: (A) CD61+ PS+, (B) CD45+ PS+ and (C) CD3+ PS+ EV number in healthy control dogs (*n* = 13) and dogs with cancer (*n* = 15). The boxes represent the 25th and 75th percentiles and the central lines represent the median values. The whiskers represent the 10th and 90th percentiles. Asterisks indicate a statistically significant difference *p* < 0.05 between control dogs and dogs with cancer.

## 4. Discussion

As submicron particles, EVs are believed to play a role as messengers in cell-to-cell communication. Their quantity is increased in many pathological conditions, from inflammation to cancer [18,19]. A mouse model of cardiac injury showed a higher number of EVs released from cardiomyocytes with specific cargo (isoform of glycogen phosphorylase), which could be an early and sensitive marker of heart injuries [20]. In plasma, elevated numbers of PEVs, leukocyte EVs, or endothelium-derived EVs in humans have been reported in systemic lupus erythematosus, acute myocardial infarction, renal failure, atherosclerosis, and infections. Some papers have revealed increased numbers of PEVs in colorectal and mammary cancer and confirm their modulating effect on the cancer cells [21]. However, to our knowledge no studies regarding EVs in dogs with cancer have been performed.

Recently, many works have focused on methods of EV isolation, identification, and enumeration, but there have been only a few regarding veterinary medicine. It has also been discussed how pre-analytical steps affect EV number. To avoid the effects of vascular damage by venipuncture and platelet activation, the standard procedure involves first discarding 1–2 milliliters of blood. The gauge needle should also be large enough in diameter to prevent in vitro hemolysis, platelet activation, and formation of erythrocyte-derived EVs [22].

Jayachandran et al. examined the influence of anticoagulants EDTA (ethylenediaminetetraacetic acid), Na citrate, ACD (acid citrate dextrose), and sodium heparin on EV (PEVs and EEVs) quantity in human blood. They revealed that the numbers of all examined EVs were significantly lower in the samples containing calcium chelating anticoagulants compared to protease inhibitors. This could result from in vitro formation of PEVs but not EEVs in blood during storage, or their quicker loss than in heparinized blood [14].

The next widely discussed aspect of EV isolation is the blood centrifugation after collection. The speed of centrifugation impacts the size of the analyzed EVs, and for EXSM isolation, ultracentrifugation at 100,000–200,000× *g* is recommended [23]. There are two common techniques of centrifugation to obtain ECTSMs. Nielsen et al. proposed serial centrifugation at low speeds (10 minutes at 1800× *g*; 15 minutes at 3000× *g*; 5 minutes at 3000× *g*) [17]. Lacroix et al. examined two procedures of centrifugation: (a) 2 × 15 minutes at 2500× *g* or (b) one centrifugation of 15 minutes at 1500× *g* and a second centrifugation for 2 minutes at 13,000× *g*. The number of EVs was significantly higher in the second method of centrifugation, but in their conclusions, they recommended the first variant as good enough and more practical in the majority of diagnostic laboratories [22]. The first Lacroix centrifuge method was used in evaluation of PS+ EVs in dogs by Kidd et al. [24] In our study, we modified this protocol for a longer time of blood storage and for better elimination of cells and cellular debris from PFP.

Helmond et al. measured PEV numbers in healthy dogs in platelet-poor plasma (PPP), microparticle-free plasma (MPF), and microparticle-enriched plasma (MPEP), and PEV medians were 505, 108, and 1850/µL, respectively [25]. The median PPP values in the study of Helmond et al. were higher than in our study, which could be caused by different parameters of EV estimation, including fresh citrate plasma and the parameters of centrifugation. Helmond et al. noticed that routine estimation of EVs in fresh samples in veterinary practice may be limited. Kidd et al. measured procoagulant microparticles with PS+ expression in healthy dogs and in dogs with IMHA immune-mediated hemolytic anemia) [24]. The same parameters of centrifugation were used to prepare PPP as Helmond’s, and the PS+ EV median was 251,000/µL in healthy dogs and 361,990/ µL in IMHA dogs. Attained values were higher than in Helmond’s work, but this was due to the examination of one parameter only (PS+) on the EV membrane [25].

It has been often discussed which membrane surface marker is the best for EVs. McEntire and colleagues measured PS+ EVs in RBC storage concentrates. The purpose of the study was to estimate EV number using annexin V and two new phosphatidylserine markers that have not been used in veterinary practice, but the origin of the EVs was not examined [26]. Lactadherin binds PS-enriched cell and microparticle surfaces in a receptor-independent manner, and bio-maleimide conjugates non-specifically with cysteine residues in cell/ EV membranes. The authors concluded that the counts of AnV+ and bio-maleimide+ EVs were comparable, but the lactadherin+ EV number was higher, which seems to have been an effect of higher sensitivity to PS than to AnV [26].

In humans, PEV count is higher in healthy women than in men, and varies depending on the phase of the menstrual cycle [25,27]. In dogs, Helmond et al. similarly observed higher numbers of PEVs in females (median 711/µL) compared to males (median 413/µL) [25]. In our study, the number of PEVs in females from the control group was significantly lower than in the examined group (control median 160/µL; cancer median: 404/µL). However, the numbers of males were too low in the control and examined groups to make a comparison between the sexes.

The primary aim of this study was to detect EVs using annexin V and specific cell surface antigens, and to compare their number in healthy dogs and in dogs with cancer. Based on SSC Megamix Plus beads, in our study, we analyzed only EVs in the range 0.24 to 0.5 µm on the SSC-H scale, which is the size of ECTSMs or large EVs. It is well documented that in most cases of cancer in humans, PEV count is increased. We found that the CD61+ PS+ EV number in dogs with cancer was higher when compared to healthy controls. The results obtained in our work confirm similar observation in humans made by Baran et al. in patients with different stages of gastric cancer and in other types of neoplasia [28,29].

It is interesting that CD61 expression on EVs was initially considered to indicate rather megakaryocyte- than platelet-derived EVs [30]. Flumenhaft and colleagues suggested differentiation of this EV populations based on co-expression of CD62P and the membrane protein C-type lectin-like receptor-2 (CLEC-2) for PEVs. CLEC-2 expression and lack of GPVI is characteristic for megakaryocyte (Mks)-derived EVs (MKEVs) [30,31]. CD62P is one of the most commonly used markers for assessing platelet activation. CLEC-2 mediates the production of thrombopoetin and other factors in Mks, but is also expressed on platelets [31,32,33]. Currently, CD61 has been approved as a marker of PEVs in humans and other species [2,10,34,35].

Expression of CD45, a common leukocyte antigen, was observed on megakaryocytes and only on 0.34 % of mature platelets in healthy individuals [35,36]. In this study, we have described for the first time EVs with CD61+ and CD45+ co-expression. They could originate from Mks, however, it would require additional study using the abovementioned antibodies to confirm this hypothesis. In our study, the number of CD45+ CD61 + EVs was relevantly lower than PEVs in both examined groups. It is also interesting that the CD61+ CD45+ EV number was not significantly higher in dogs with cancer, which could suggest that they may not be so important in cancer cell-to-cell communication, but this needs further investigation.

There was no correlation between PLT and PEV numbers in healthy dogs and dogs with neoplasia. PLT counts were significantly higher in the examined group, but only three dogs had PLT numbers above reference values. The reason for the lack of correlation may be the cancer heterogeneity of the examined group. It has also been confirmed that PEV numbers in many types of neoplasia increase, but the PLT count is elevated in only a few types of cancer in humans [28].

One of the limiting factors in EV evaluation is their submicron size, and, as a consequence, a merged number of their surface antigen copies from parental cell. The diversity of antigen density on cell surfaces depends on the antigen and cell origin. In humans, there are 210 thousand copies (tc) of CD21 on B lymphocytes, compared to only 18 tc of CD19, but on T cells, CD3 has 124 tc and CD28 has 20 tc [37,38]. There have been a few works in humans describing EVs with CD3 and CD21 expression and their influence on cells. Blanchard et al. showed that activation of T cells from healthy humans and Jurkat T cells via TCR receptor increased the production of EVs bearing CD3, whereas the mitogens PMA and ionomycin did not induce their formation [39]. Interestingly, these EVs did not express CD45, which suggests that this type of EV was originally from the endocytic compartment [39]. Other authors have shown that CD3+ EVs from healthy patients had lower levels of PD-L1, COX-2, and CD15s compared to patients with neck squamous cell carcinoma [40]. Furthermore, they observed a higher percentage of CD3+ EVs with co-expression of immunoregulatory proteins PD-L1, CD15s, CTLA4, and COX-2 in patients with stage III or IV compared to stage I and II, confirming the usefulness of EVs as promising non-invasive cancer biomarkers and their immunomodulation potential [40]. In our study, an increased number of CD3+ EVs may have resulted from lymphocyte activation and anticancer response, but this requires further research. 

In this study, we examined a heterogenic subpopulation of EV microparticles. To determine the number and diameter of a smaller EV subpopulation, exosomes, researchers need more accurate techniques: nanoparticle tracking analysis (NTA) or transmission electron microscopy (TEM). TEM can be used to analyze single ecto- and exosomes. Both of these imagining techniques are considered reference methods for exosome research, as published in a statement of the International Society for Extracellular Vesicles in 2018 [41,42].

The main limitation of our study was the heterogeneity of the group. In this pilot study, we decided to examine a heterogeneous group of cancer patients, in order to study more cancer types. Analysis of a higher number of dogs with one type of neoplasia (in our area of interest are dogs with lymphoma and osteosarcoma) would increase the statistical power and might reveal statistical significance where it can be only suspected in this study, such as the increases in CD45 + EVs and CD21 + EVs in dogs with cancer. Additionally, as shown by McEntire et al., lactadherin as a PS marker can be used in future tets [26].

## 5. Conclusions

Increasing attention turns to the study of EVs’ role in a variety of pathological conditions, using them as biomarkers or as potential therapeutic targets. Their role as a potential new class of therapeutic agents has also been discussed. Encapsulated oncolytic adenoviruses in EVs in a mouse model efficiently targeted and destroyed cancer cells, and understanding of the mechanism of the reaction may lead to creating more individualized and effective cancer therapies [43,44]. 

Our study demonstrated that the number of PEVs and EVs from leukocytes and EVs derived from T cells was higher in dogs with cancer compared to healthy controls. The estimation of PEVs and leukocyte-derived EV counts may provide a useful biological marker in dogs with cancer. In veterinary medicine, further evaluation of EV specificity and their influence on target cells in different conditions should be performed to confirm their diagnostic and therapeutic potential. Dogs and their owners share the same living environment, and are thus exposed to the same environmental factors. The higher frequency of certain types of canine neoplasia (lymphoma, osteosarcoma) and their similarities in biological behavior, histological, and molecular profile and response to chemotherapy to human cancers means that dogs in some spontaneous types of neoplasia could be considered a valuable cancer animal model.

## Figures and Tables

**Figure 1 animals-09-00575-f001:**
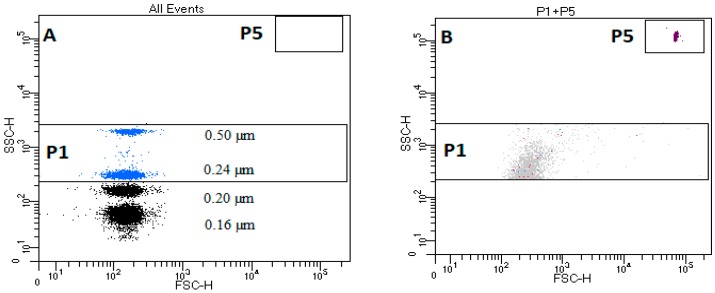
Flow cytometry gates and thresholds for characterizing blood microvesicles in dogs. (**A**) Characteristic dot plot of Megamix SSC beads; P1—rectangular region for extracellular vesicles (EVs) located between beads 0.2–0.5 µm on the y-axis H-SSC; P5 region—rectangular region of TruCountTM beads. (**B**) Characteristic dot plot of TruCountTM beads in filtered DPBS. (Dulbeccos’ PBS).

**Figure 2 animals-09-00575-f002:**
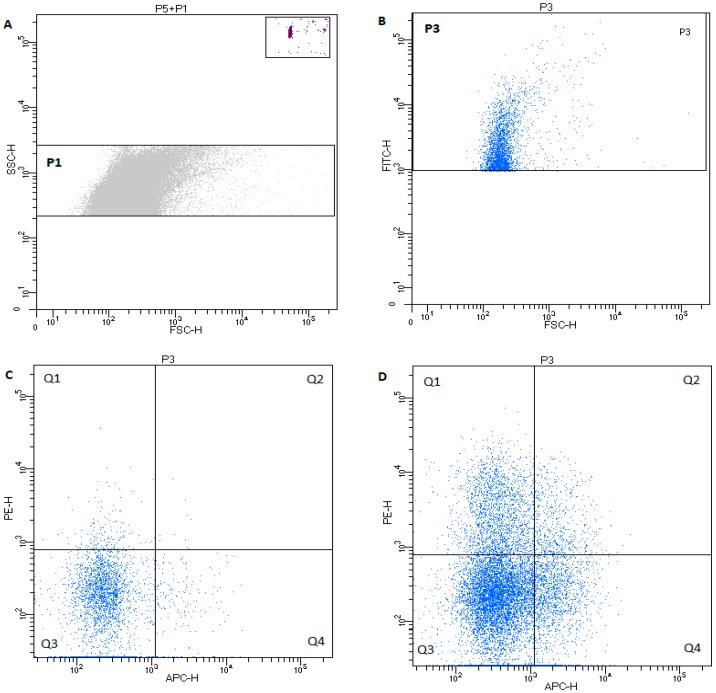
Characteristic of dot plot of microvesicle regions in healthy dog. (**A**) P1—rectangular region of EVs 0.24–0.5µm on the SSCH scale; and P5—rectangular region of TruCountTM beads. (**B**) Characteristic of dot plot events from P1 EV region labeled with annexin V (PS+)—region P3. (**C**) Microparticles with CD61 (PE) and/or CD45 (APC) expression in healthy dogs from the P3 region. Events in quadrant 1 (Q1) are positive for CD61 and PS, and quadrant 2 (Q2) events are positive for CD61, CD45, and PS. The microparticles in (Q4) are CD45+ PS+. Events in quadrant 3 (Q3) represent debris, machine noise, and microparticles that did not bind either fluorescent label. (**D**) Exemplary dot plot of annexin-V-positive microparticles with CD61 (PE) and/or CD45 (APC) expression in dogs with cancer.

**Figure 3 animals-09-00575-f003:**
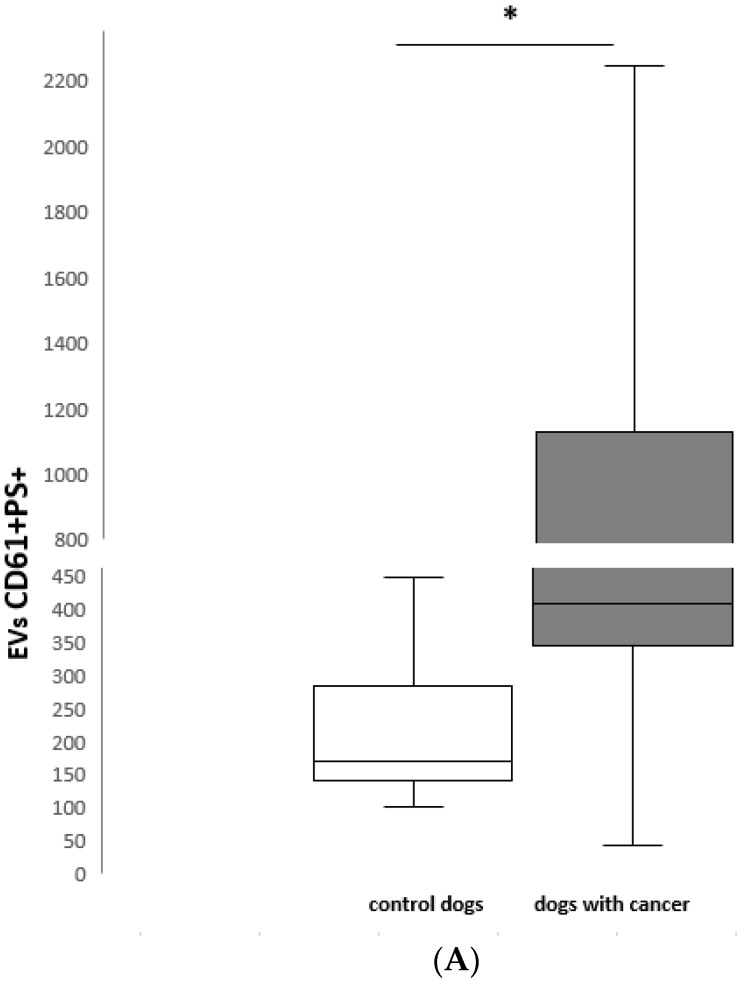
(**A**) CD61 + PS + EVs number/µL in healthy control dogs (*n* = 13) and dogs with cancer (*n* = 15); (**B**) CD45 + PS + EVs number/µL in healthy control dogs (*n* = 13) and dogs with cancer (*n* = 15); (**C**) CD3 +PS + EVs number/µL in healthy control dogs (*n* = 13) and dogs with cancer (*n* = 15). * *p* < 0.05 between control dogs and dogs with cancer.

**Table 1 animals-09-00575-t001:** Hematological variables in control dogs and in dogs with cancer.

Variable	Control Dogs (*n* = 13) Nedian (Range)	Dogs with Cancer (*n* = 14) Median (Range)
PCV L/L	0.39 (0.37–0.58)	0.37 (0.31–0.49)
RBC × T/L	6.2 (5.3–8.5)	5.9 (4.8–7.4)
MCV fL	64 (62–71)	61 (60–77.8)
MCHC g/L	371 (309–421)	330 (243–370)
WBC × G/L	7.1 (5.8–9.8)	15 (6.7–31)
Platelet count × G/L	254 (182–371)	422 (165–598) *

Abbreviations: PCV, packed cell volume; RBC, red blood cells; MCV, mean cell volume; MCHC, mean corpuscular hemoglobin concentration; WBC, white blood cells; PLT, platelets; * *p* < 0.05.

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
