# Peer review of "Extracellular Vesicles in the Blood of Dogs with Cancer—A Preliminary Study"

_animals, 2019, doi:10.3390/ani9080575_

Round 1

Reviewer 1 Report

Broad comments

All sections: Introduction, Material and methods, Results and Discussion are well and interestingly written. The weakness of the presented work is a relatively small research group (especially of sick dogs) which makes it impossible to draw detailed conclusions regarding the potential use of EVs in diagnostics and prognosis. There is also no possibility to draw conclusions about the significance of the number of EVs in different types of cancer in dogs. However, these are preliminary studies and as such are well prepared and presented. Trends are clearly indicated (more EVs in dogs with tumors than in healthy dogs). As a work that presents the results of preliminary research, it is also a great description and analysis of the methodology allowing this type of research - and this analysis is the strongest part of the article. It indicates the possibility of an innovative research direction in veterinary oncology and provides proven information on how to conduct such research.

I suggest a slight change in the design of the manuscript. Information on the number of a different type of cancer samples from sick dogs should be provided in M&M section not in Results.

Due to the pioneering nature of the research, it is impossible to conduct a broad discussion of the results obtained. In the opinion of the reviewer, however, one should at least try to correlate the results obtained to the results of hematological blood tests. In the study, the authors found a clearly higher number of platelets in cancer patients. In this group of patients, a higher number of PEVs was also found. Is this related? It would be worth adding speculation on this subject in the discussion.

Specific comments

Information on the number of a different type of cancer samples from sick dogs should be provided in M&M section not Results.

2.3. Statistical analysis

Page: 5 Line: 163 P<0,05 - change "," into "."

Author Response

 Thank You for all the suggestions. We have revised the article as it had been suggested.

The weakness of the presented work is a relatively small research group (especially of sick dogs) which makes it impossible to draw detailed conclusions regarding the potential use of EVs in diagnostics and prognosis. There is also no possibility to draw conclusions about the significance of the number of EVs in different types of cancer in dogs.

Replay: We agree that size of the examined group is small. In our study admission criteria for examined group were restricted. In examined group only dogs with newly diagnosed cancer but without medical treatment during last two weeks were included. Also plasma collected form patients was stored for only 4 months.

There is also no possibility to draw conclusions about the significance of the number of EVs in different types of cancer in dogs.

Replay: These studies are preliminary and we avoided to conclude with what phenotype EVs are crucial in various types of cancer. Our work confirms that number of: platelets, leukocytes and T cells derived EVs is higher in dogs with neoplasia. The next step of our investigation is to asses number of PEVs and other EVs for the particular type of cancer in different clinical stages, which may poses them as a useful biological marker.

I suggest a slight change in the design of the manuscript. Information on the number of a different type of cancer samples from sick dogs should be provided in M&M section not in Results.

Replay: We change that information into M&M section: 2.1.  paragraph –Animals.

In the study, the authors found a clearly higher number of platelets in cancer patients. In this group of patients, a higher number of PEVs was also found. Is this related? It would be worth adding speculation on this subject in the discussion.

Replay: We have considered that remarks as a good point that need to be discussed and, thus we have added some comments in discussion section.

Page: 5 Line: 163 P<0,05 - change "," into "."

Replay: We change that mistake.

Reviewer 2 Report

The manuscript written by Å»migrodzka et al describes an interesting approach: Isolating EVs from the blood of dogs with cancer.

1) However as the litterature about EVs is extensive and a lot of research is conducted on the use of EVs as drug delivery strategy, this aspect should be reported. Authors should at least mention that EVs have been used as delivery vehicles for chemotherapeutic agents ( Yarana Clin Cancer Research 2018), oncolytic viruses (Ran Biomaterials 2015, Garofalo Viruses 2018) thus highlighting their potential use in cancer treatment.

2)Material and Methods: how many ml of blood have been used for EV-isolation?

Which was the dog´s weight? Did authors collected the same amount of blood for each dog?

3) A proper characterisation is needed: NTA and TEM analyses have to be carried out

4) A real advantage of using EVs from dogs is not clear. Authors must state it in order to highlight the novelty and the strenghts point of their work

5)Did the authors perform pharmacokinetic studies? How is the distribution of these EVs?

Author Response

Thank You for all the suggestions and favourable comments. We have revised as it had been suggested.

However as the litterature about EVs is extensive and a lot of research is conducted on the use of EVs as drug delivery strategy, this aspect should be reported. Authors should at least mention that EVs have been used as delivery vehicles for chemotherapeutic agents ( Yarana Clin Cancer Research 2018), oncolytic viruses (Ran Biomaterials 2015, Garofalo Viruses 2018) thus highlighting their potential use in cancer treatment.

Reply: As the reviewer had recommended, we have upgraded those information in the Discussion and Conclusions chapters.

Material and Methods: how many ml of blood have been used for EV-isolation?

Which was the dog´s weight? Did authors collected the same amount of blood for each dog?

Replay: We always collected plasma from the same amount of blood and we added that information in the manuscript.

A proper characterization is needed: NTA and TEM analyses have to be carried out.

Replay: We agree with reviewer, but this work had limitation because of relatively small amount of collected blood. Also NTA and TMEM technique are relatively more expensive compared to flow cytometry. We decided using FC method as “more affordable” in veterinary medicine in future diagnostic procedures.  

A real advantage of using EVs from dogs is not clear. Authors must state it in order to highlight the novelty and the strengths point of their work

Replay: The strong point of our work is that for the first time EVs number and their phenotype in plasma was estimated in dogs with neoplasia. Further research is crucial to show their role as a useful biological marker in dogs with variety types of neoplasia.

Round 2

Reviewer 2 Report

Authors improved the manuscript.

However my concern is still related to the missing EV-characterisation according to MISEV2018, as reccomended by the International Society of EVs.

I suggest to discuss this aspect if authors are not willong to add these data into the present manuscript, in order to make this work appealing and trustable for EV-society

Author Response

Thank You for Your suggestion. I've added the following information to the discussion section of the manuscript.

In this study we examined heterogenic subpopulation of EVs – microparticles. To determine the number and diameter of smaller EVs subpopulation- exosomes, researchers need more accurate technics: Nanoparticle Tracking Analysis (NTA) or transmission electron microscopy (TEM). TEM is used to analyze single ecto- and exosomes. Both of these imagining technics are considered as reference methods for exosome research which is published in statement of the International Society for Extracellular Vesicles in 2018 [41,42].